

# Establishment of a regional precipitable water vapor model based on the combination of GNSS and ECMWF data

**Yibin Yao[1, 2], Xingyu Xu[1], Yufeng Hu[1,*]**

[1] *School of Geodesy and Geomatics, Wuhan University, 129 Luoyu Road, Wuhan, 430079, China. E-mail: ybyao@whu.edu.cn*

[2] *Key Laboratory of Geospace Environment and Geodesy, Ministry of Education, Wuhan University, 129 Luoyu Road, Wuhan, 430079, China;*

* Correspondence: yfhu@whu.edu.cn

**Abstract:** Water vapor is the engine of the weather. Owing to its large latent energy, the phase changes of water vapor significantly affect the vertical stability, structure and energy balance of the atmosphere. Many techniques are used for measuring the water vapor in the atmosphere such as radiosondes, Global Navigation Satellite System (GNSS) and water vapor radiometer (WVR). In addition, the method that uses European Centre for Medium-range Weather Forecasts (ECMWF) data is an important method for studying the variations in precipitable water vapor (PWV). This paper used both GNSS PWV and ECMWF PWV to establish a city-level local PWV fusion model using a Gaussian Processes method. The results indicate that by integrating the precipitable water vapor obtained from GNSS and ECMWF data, the accuracy of fusion PWV is improved by 1.89 mm in active tropospheric conditions and 2.61 mm in quiescent tropospheric conditions compared with ECMWF-PWV, reaching 3.87 mm and 3.97 mm, respectively. Furthermore, the proposed fusion model is used to study the spatial and temporal distribution of PWV in Hong Kong. It is found that the accumulation of PWV corresponds to monsoon and rainfall events.

**Keywords:** Precipitable water vapor (PWV); Global Navigation Satellite System (GNSS), European Centre for Medium-range Weather Forecasts (ECMWF), Fusion model

## 1. Introduction

Water vapor is a highly variable component in the atmosphere and plays a key role in many atmospheric processes. Accurate measurement of water vapor is vital for improving the predictability of regional precipitation, weather and visibility, especially for a highly moist metropolis such as Hong Kong (Chen and Liu, 2014). Many techniques are used for water vapor measurement in the atmosphere, such as radiosondes, ground- or space-based water vapor radiometers, Global Navigation Satellite System (GNSS) and other meteorological methods.

Radiosonde can accurately measure the water vapor, but its high operating cost restricts its applications in short-term weather forecasting. Its temporal and spatial resolution is quite poor (Guerova, 2003), usually with a 12-h observation interval. Since GNSS meteorology was first proposed by Bevis et al. (1992) as an approach for sounding the atmospheric water vapor by using ground-based receivers, extensive investigations based on batch processing have been conducted in the past two decades (Rocken et al., 1997; Ohtani and Naito, 2000; Hagemann et al., 2003; Braun., 2004; Gendt et al., 2004). GNSS has several significant advantages, including a low operating cost, all-weather availability, and high spatiotemporal resolution (Lu et al., 2015).



Various studies have proven that GNSS can provide accurate water vapor estimates comparable to the measurements obtained from meteorological sensors in both post-processing and near-real-time modes (Gendt et al., 2004; Haan et al., 2004; Gutman et al., 2004; Elgered et al., 2005; Nilsson and Elgered, 2008). However, the uneven distribution of ground GNSS stations has resulted in limited PWV coverage in marine regions and other remote areas. The ECMWF produces the highest level of short-term numerical weather forecast in the world and can provide global water vapor data 4 times a day (Annamalai et al., 1999; Huang et al., 2006; Renfrew et al.2002; Bromwich et al., 2004). Because of the consistency and homogenous spatial coverage of ECMWF data, they play an increasingly important role in regional weather forecasting and are being increasingly studied by scholars (Flentje et al., 2007; Ye et al., 2007; Zhang et al., 2009; Bock et al., 2010). The high-precision ECMWF reanalysis product, ERA-Interim, does not assimilate ground-based GNSS observations and extends back to 1979 (Dee et al., 2011), thereby maintaining good continuity.

Over the last several years, the assimilation of GNSS PWV into mesoscale numerical prediction models have been widely investigated (e.g., Guerova et al., 2004; Vedel et al., 2004; Nakamura et al., 2004; Smith et al., 2007; Secoetal, 2009). Additional applications are concerned with validating PWV reanalysis products with GNSS observations (Vey et al., 2010). Although each water vapor measurement method has its advantages and disadvantages, the data are usually used alone. Only a few efforts have been devoted to investigating the modeling of multi-source precipitable water vapor data. In this paper, using both ECMWF and GNSS data, we aim to establish a local PWV fusion model. It is expected to obtain PWV field with higher accuracy and higher horizontal resolution, which is more suitable for weather analysis. The fusion is conducted with Gaussian Processes, and the results of using multisource data are validated with the radiosonde data and ground–based GNSS data. In addition, the spatial and temporal distribution of PWV in Hong Kong is analyzed based on the proposed fusion model, which is a preliminary exploration of model application.

## 2. Materials and Methods

### 2.1. Data description and processing strategy

2.1.1. GNSS PWV

GNSS methods dedicated to estimating the PWV and are now well developed and commonly applied (Rocken et al., 1997; Ohtani and Naito, 2000; Hagemann et al., 2003; Braun, 2004). This technique is based on estimating the tropospheric delay by using a Global Navigation Satellite System (GNSS) with a combination of surface pressure and temperature.

The study is based on ground-based GNSS measurements of PWV from the Hong Kong Satellite Positioning Reference Station Network (SatRef).

Figure 1 shows the location map of the SatRef network continuously operating reference stations (CORS), and the radiosonde station is marked with a five-pointed star.

The Hong Kong SatRef network consists of 15 continuously operating reference stations equipped with Leica GNSS receivers and antennas (Figure 1). Each station (except T430) is equipped with an automatic meteorological device to record the temperature, pressure and relative humidity. With these data, the hydrostatic components of the tropospheric delay can be accurately estimated. The mean horizontal





distance between stations is approximately 10 km, and the ellipsoidal heights of the 15
stations are within 350 m. GNSS observation data from the SatRef Network are
processed by the precise point positioning (PPP) module in the Bernese 5.0 software
(Astronomical Institute of the University of Bern, Bern, Switzerland) (Dach et al.,
97   2007).
98       Once the zenith troposphere delay is obtained through the PPP, the precipitable
water vapor can be calculated. A brief description of the computation procedure of the
estimation of PWV is given below:
The temperature, pressure and relative humidity recorded by meteorological
devices at 14 tracking stations are used to calculate Zenith Hydrostatic Delay (ZHD).
Therefore, ZHD can be calculated from the surface pressure $P_s$ (mbar), latitude $\varphi$
(radians) and ellipsoidal height $H_s$ (km) using the equation given by Saastamoinen et al
105  (1972):

$$ZHD = 2.2768 \times P_s / (1 - 0.00266 \cos 2\varphi - 0.00028h) , \qquad (1)$$

ZWD is obtained by subtracting the ZHD from the ZTD. Subsequently, the
precipitable water vapor can be calculated from the Zenith wet delay (ZWD) and
dimensionless proportional constant $\Pi$ . ZWD is converted into PWV using the
following expression (Wang et al., 2005):
$$PWV = \Pi \times ZWD , \qquad (2)$$

$$\Pi = \frac{10^6}{\rho_w R_v [(k_3 / T_m) + k_2']} , \qquad (3)$$

where $k_2' = 16.529 k \cdot mb^{-1}, k_3 = 3.7339 \times 10^5 k \cdot mb^{-1}$ , $T_m$ is the weighted mean temperature
of the atmosphere, $\rho$ is the density of water, and $R_v$ is the gas constant for the water
vapor. $T_m$ (Kelvin) is given by the GPT2w model (Böhm et al., 2014).

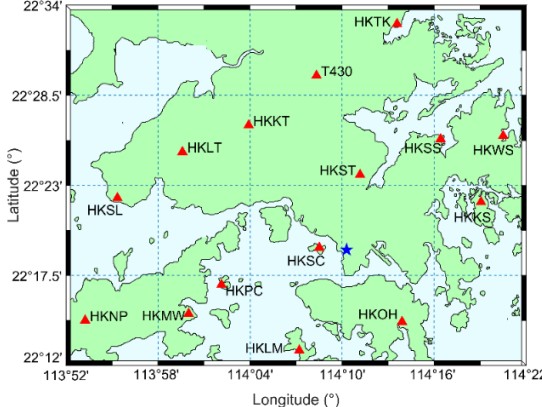

**Figure 1.** The SatRef network in Hong Kong
2.1.2. ECMWF PWV
To compare PWV measurements from the latest reanalysis product with
radiosondes, surface PWV data of Hong Kong from ERA-Interim (Dee et al., 2011),
covering the time period July–August 2015, are considered. In order to incorporate
more dense data into the model, data with a spatial resolution of 0.125 °×0.125 ° in
longitude and latitude and a temporal resolution of 6 h (at 0, 6, 12 and 18) have been
retrieved from the ECMWF archive. In general, data from 25 grid points over Hong
Kong are used, covering 22.125 °–22.625 °N and 113.875 °–114.375 °E.
It is expected that by integrating the water vapor data obtained from other
observations such as GNSS and WVR into the ECMWF, the ECMWF can improve its
capability in short-term severe weather prediction. This is precisely the research
motivation of this paper.
2.1.3. Radiosonde PWV
The only radiosonde station in the Hong Kong region is situated at the King's Park
(22.32 °N, 114.17 °E). Radiosonde balloons are launched twice per day by the Hong
Kong Observatory (HKO) at UTC 0:00 and 12:00 (local hour: UTC+8). The radiosonde
instrument used by the HKO is the Vaisala RS92, which claims to have a reproducibility
better than 2 %. Each radiosonde can measure meteorological parameters such as
pressure, temperature and relative humidity at various altitudes using the balloon-borne
platform. In this paper, the data from the Hong Kong radiosonde station at 0:00 and
12:00 UTC during two weeks in July and August of 2015 were selected. By using these
meteorological parameters, the PWV at the radiosonde station is calculated according
to formula (4). Since the radiosonde can measure PWV with an accuracy of a few
millimeters, PWV measurements derived from radiosonde data are often used as an
accuracy standard to evaluate the water vapor data from other independent sensors
(Niell et al., 2001; Adeyemi et al., 2012). Because of its expensive operating cost,
radiosonde data have a low temporal data rate, which limits their applications in short-
term weather forecasting.
A commonly used quantity in meteorology is the precipitable water vapor
calculated by an integration method, which is defined as
$$PWV = \frac{1}{\rho_1}\int_H^\infty \rho_w dz = \frac{1}{\rho_1 R_w}\int_H^\infty \frac{e}{T}dz$$
. (4)

Here, Rw=R/Mw, and ρ1 (the liquid water density) is chosen to be 1000 kg/m3.
*2.2. Local PWV model using multisource data*
Fitting the PWV values obtained by different methods of observation using an
appropriate model can produce a local PWV model. The method used in this paper is
polynomial fitting through a Gaussian Processes (GP) model (Xia.et al., 2008; del
Castillo et al., 2015; Colosimo et al., 2014). A GP model (the technique also known as
kriging) is a particular type of random process in which the probability distribution
function (*pdf*) associated with any process observation is normal, and the joint
probability distributions associated with any finite subset of process observations are
normal as well (Cressie, 1993; Williams et al., 2006; Forrester et al., 2008). Formally,
a GP model is defined by Eq. (5):
$$PWV(v_i) = f(v_i) + \varepsilon(\varepsilon \sim N(0, \sigma_\varepsilon^2))$$
$$f(v_i) = GP(m_{pwv}(v_i), k_{pwv}(v_i, v_j))$$
, (5)

Where $v_i = (B_i, L_i)$ and $m_{pwv}(v_i) = E[PWV(v_i)]$ is the mean function, which is used to
describe the expected *pwv* value at $v_i$. The term $\varepsilon$ accounts for the measurement error,
and is assumed to follow a normal distribution with 0 mean and $\sigma_\varepsilon^2$ variance, i.e.





$\varepsilon \sim N(0, \sigma_\varepsilon^2)$ . $k_{pwv}(v_i, v_j) = E[(PWV(v_i) - m_{pwv}(v_i))(PWV(v_j) - m_{pwv}(v_j))]$ is the covariance of
the *pwv* value at locations $v_i$ and $v_j$.
To reduce the fitting coefficients, the PWV involved in modelling have been
height(m)-reduced to the earth surface using coefficients obtained from Ref (Means,
2011). The reduction equation is as follows:
$$PWV(h) = PWV_0 e^{-h/2697} \qquad . \tag{6}$$

In this work, we used a quadratic polynomial to represent the mean function of the
GP model. The polynomial function model is expressed as follows.
$$m_{pwv}(v_i) = a_0 + a_1 B_i + a_2 L_i + a_3 B_i L_i + a_4 B_i^2 + a_5 L_i^2 (i = 1 \cdots n_1 \cdots n_2) \tag{7}$$

Where $n_1$ denotes the number of reference GNSS stations and $(n_2-n_1)$ denotes
ECMWF grid points respectively, and the subscript $i$ denotes the index of the reference
stations. $PWV_i$ is the surface precipitable water vapor at the $i$th station. $(B_i, L_i)$ are the
latitude and longitude of the station. $(a_0, a_1, \ldots \ldots a_5)$ are six fitting coefficients.
In addition, we used the squared exponential function to represent the covariance
of the GP model:
$$k_{pwv}(v_i, v_j) = \sigma_{pwv}^2 \exp(-\frac{\| v_i - v_j \|^2}{2l^2}) , \tag{8}$$

where $\| v_i - v_j \|$ is the Euclidean distance between locations vi and vj in the plane, $\sigma_{pwv}^2$
is the constant variance of the GP model and $l$ is the characteristic length-scale. In
practice, according to Eq. (8), *pwv* values that lie closer together on the plane
(regardless of where they are located) are likely to be more similar. The squared
exponential is one of the most popular choices for GP models because it yields positive
definite correlation matrices, enables the proper convergence of the statistical
estimation algorithms and can model smooth and infinitely differentiable functions
(Rasmussen et al., 2010).
The parameters $\{a_0, a_1, a_2, a_3, a_4, a_5, \sigma_\varepsilon^2, l, \sigma_{pwv}^2\}$ of the geometry model described by
Eqs. (5)- (8) are all unknown and must be estimated from the actual measurement data
$PWV(v_i)$. The fitting of GP models was implemented in this paper based on the code
developed by Rasmussen et al. (2010). Once the parameter estimation is complete, the
knowledge of the mean and covariance functions make it possible to estimate the value
of the function *pwv(v)* at any new location *v* in the plane.
To investigate the contribution of GNSS observations, this paper uses the PWV
derived from 7 CORS tracking station (HKOH, HKPC, HKST, HKSS, HKSL, HKTK,
and HKWS) with uniform distribution and uninterrupted observations. Adding the
water vapor data to ECMWF PWV reanalysis products provide a certain help to
improve their accuracy and reliability. In this paper, we consider two different
situations with active and quiescent troposphere conditions on days of year (DOYs)
201~207 and 213~219 in 2015, respectively. The weather condition on DOY201~207
is relatively active compared with those on the preceding and following days. Several
severe rainfall events occurred on these days, with the largest daily rainfall (~190 mm)
in 2015 on 22 July, indicating an accumulating phase of the troposphere ZWD. The
following days of DOY213~219, however, all happened to be sunny days. For each
case, we first determine the PWV fitting coefficients using Gaussian Processes by
inputting GNSS-PWV on several CORS stations and ECMWF-PWV at grid points, and
we then assess the performance of the PWV fusion model.





## 3. Results

In this section, verification of PWV fusion model is conducted. The precision of the ECMWF reanalysis products is first evaluated. Two case studies concerning active and quiescent troposphere conditions are analyzed to assess the performance of the PWV fusion model, which is also used to study the spatial and temporal variation of PWV over Hong Kong.

To verify the contribution of GNSS material, the PWV data derived from the radiosonde station and some of the CORS stations are utilized to evaluate the precision of the calculated PWV values. The PWV accuracy for each site is expressed as the bias and RMS error of the difference between the calculated PWV and the reference PWV. The optimum criterion is defined as follows:

$$bias = \frac{1}{N}\sum_{i=1}^{N}\left(PWV_i - ZTD_i^{\text{Re ference}}\right),$$

$$RMS = \sqrt{\frac{1}{N}\sum_{i=1}^{N}\left(PWV_i - ZTD_i^{\text{Re ference}}\right)^2} \qquad (9)$$

The integrated precipitable water vapor at the radiosonde station and several GNSS-derived PWV are used to assess the accuracy of the ECMWF PWV estimates and the fusion PWV values. Due to the inconsistent locations of the grid points of ECMWF products and the radiosonde station, the ECMWF-PWV in each grid are interpolated to the radiosonde station before comparison. However, because of the complex topography with large undulations in Hong Kong, the elevation differences between the radiosonde station and the ECMWF grid points are significant, and the extracted PWV values of the ECMWF products could not be suitable for any reliable comparison with the radiosonde PWV values. Therefore, to overcome the bias between the datasets due to elevation differences, the PWV from the ECMWF are reduced to the height of the radiosonde station using the exponential function illustrated in Eq. (6).

*3.1. Case study 1: active troposphere condition*

The local water vapor fusion modeling is performed at 0:00 and 12:00 UTC on 7 consecutive days (DOY201-207) in July. Firstly, with 25 ECMWF grids and 7-CORS-station network configurations as data sources, the PWV fitting coefficients are determined through Gaussian Processes for machine learning; using these coefficients, the PWV values at the radiosonde station are obtained after height reduction. As an independent external reference, the ZTD derived from radiosonde and GNSS data processing at CORS stations that are not involved in the modeling are used to assess the precipitable water vapor fusion model.

Intercomparisons have been conducted between the techniques for PWV time series measurements. The deviations of the PWV residuals between the radiosonde and calculated PWV (ECMWF and fusion model) are presented in Figure 2, and the mean bias and RMS information are shown in Table 1.


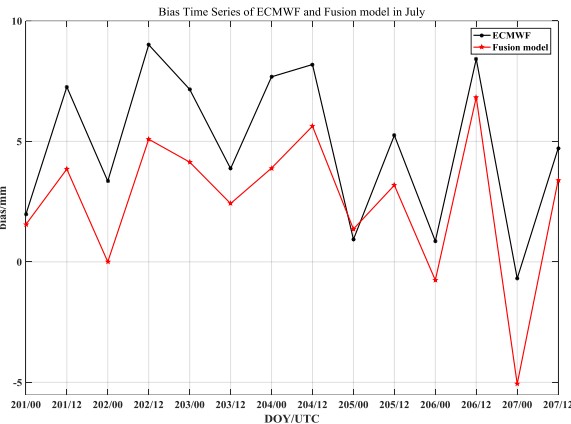

245

**Figure 2.** Bias of ECMWF and Fusion model in July

247  **Table 1** External precision of PWV fusion models versus radiosonde in July

|  | ECMWF | fusion model |
|---|---|---|
| Mean bias(mm) | 4.86 | 2.54 |
| RMS(mm) | 5.76 | 3.87 |

248

During DOY201~207, there are generally positive bias. Compared with radiosonde-PWV, the bias of ECMWF-PWV fluctuates from -0.69 mm to 9.01 mm. The highest value appears at DOY202 UTC 12:00, after when the largest daily rainfall (~190 mm) occurred on DOY203 in Hong Kong. While mean bias of the ECMWF calculated PWV is 4.86 mm, and the mean RMS reaches 5.76 mm, which is unreliable in a period of heavy rainfall. The poor accuracy is related to the imprecision of the ECMWF PWV reanalysis product processing itself. In addition, the rainstorm weather makes the meteorological material relatively inaccurate, especially in active troposphere conditions.

Unlike the ECMWF estimations, the precision of the PWV fusion model is quite reliable. The bias of PWV derived from fusion model fluctuates from -5.07 mm to 6.93 mm, the mean bias is 2.54 mm and the RMS is 3.87 mm, which is more precise than the ECMWF products. This might be because the CORS stations involved provided more abundant water vapor information. The larger bias of fusion PWV at DOY 207 UTC 0:00 might be attributed to the strong horizontal heterogeneity. The inverted atmospheric cone tens of kms wide observed by GNSS and line profile observed by the radiosonde might do not match at that moment. As shown in Table 1, by introducing GNSS data, the accuracy of PWV values calculated by the fusion model is improved by 1.89 mm from the perspective of RMS relative to the previous ECMWF products. Furthermore, the fusion model maintains high external precision and stability in active tropospheric conditions.

The PWV derived by the fusion model and GNSS observations at CORS stations that are not involved in modeling are also compared, and the average statistical results of the CORS network are presented in Table 2. In addition, a typical PWV difference



between ECMWF PWV and GNSS PWV at CORS station locations at UTC12:00 on
DOY 203 is presented in Figure 3 (left), with the difference of the fusion model shown
in Figure 3 (right). This example in Figure 3 shows that the PWVs derived from the
fusion model are more consistent with the PWV solved by GNSS.

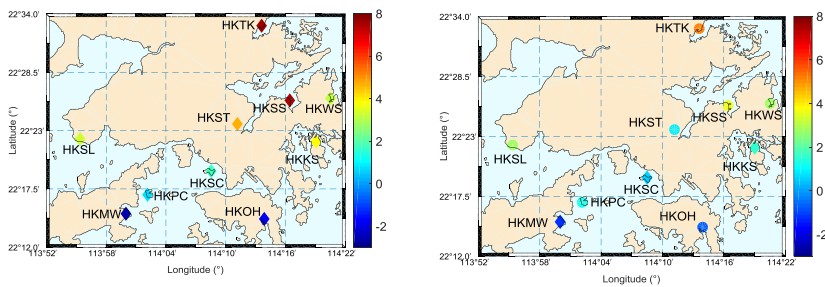


**Figure 3.** Differences (mm) between PWVs by ECMWF (left) and fusion model (right)

versus GNSS processing on the day 203 UTC 12:00

**Table 2** External precision of PWV fusion models versus CORS stations in July

| Doy | Rainfall (mm) | Hour | Mean bias(mm) | RMS(mm) |
|-----|---------------|------|---------------|---------|
| 201 | 46.2 | 0 | 1.61 | 2.24 |
| 201 |  | 12 | 4.11 | 4.30 |
| 202 | 51.2 | 0 | 3.60 | 3.91 |
| 202 |  | 12 | 4.90 | 4.96 |
| 203 | 191.3 | 0 | 5.09 | 5.78 |
| 203 |  | 12 | 0.30 | 1.16 |
| 204 | 45.0 | 0 | 3.04 | 3.11 |
| 204 |  | 12 | 3.64 | 3.91 |
| 205 | 5.7 | 0 | 1.51 | 1.70 |
| 205 |  | 12 | 2.52 | 2.65 |
| 206 | 9.6 | 0 | 3.11 | 3.20 |
| 206 |  | 12 | 2.61 | 2.68 |
| 207 | 24.9 | 0 | 6.15 | 6.21 |
| 207 |  | 12 | 2.01 | 2.27 |
| mean |  |  | 3.16 | 3.44 |


According to Table 2, using the PWV from CORS stations around Hong Kong as
a reference, the PWV obtained from the fusion model that includes data from 7 CORS
stations is more accurate, with a mean bias of 3.16 mm and a mean RMS of 3.44 mm.
These improvements indicate that adding GNSS water vapor data to ECMWF PWV
reanalysis products helps improve the accuracy and reliability of PWV.
*3.2. Case study 2: quiescent troposphere condition*
The second case study describes a stable troposphere period during 7 consecutive
days (DOY213-219) in August, when the daily sunshine duration reaches 5.7 h ~ 11.4





h. Similarly, the local precipitable water vapor fusion modeling is performed at 0:00
and 12:00 UTC each day. The deviations of the PWV residuals between radiosonde and
PWV calculated by ECMWF and fusion model with 7-CORS-station network
configurations are presented in Figure 4, and the mean bias and RMS information are
shown in Table 3.

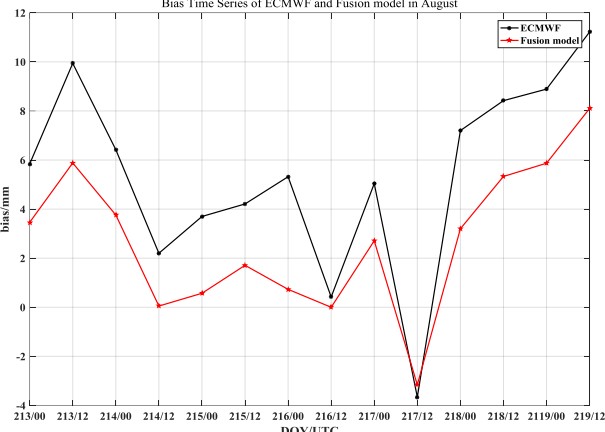


**Figure 4.** Bias versus radiosonde of ECMWF and Fusion model in August
**Table 3** External precision of PWV fusion models versus radiosonde in August

|  | ECMWF | fusion model |
|---|---|---|
| Mean bias(mm) | 5.37 | 2.73 |
| RMS(mm) | 6.58 | 3.97 |


During DOY213~219, except at DOY 217 UTC 12:00, the overall bias still appears
to be positive. Compared with radiosonde-PWV, the bias of ECMWF-PWV fluctuates
from -3.66 mm to 11.22 mm. While mean bias of the calculated PWV is 5.37 mm, and
the mean RMS reaches 6.58 mm, which is quite unreliable in the fair-weather period.
In addition, the bias shows an upward trend during DOY 218~219, especially on 219/12,
when the bias of ECMWF-PWV exceeds 10 mm, which may because that the precision
of ECMWF data decreased these days.
With the fusion PWV model, the bias fluctuates from -3.16~8.10 mm, with a mean
bias of 2.73 mm and a mean RMS of 3.97 mm. Compared to the ECMWF estimations,
the results for the calculated PWV are considerably more reliable, showing a 2.61 mm
RMS improvement. Therefore, introducing GNSS observations into the meteorological
reanalysis product has a stabilizing effect on the PWV fusion model, with a precision
improvement of approximately 3 mm relative to the previous ECMWF products.
Similar to the processing and strategies in the first case study, the PWV derived
from GNSS data processing on CORS stations that are not involved in the modeling are
used to assess the PWV fusion model. Table 4 presents the mean precision over the
inspection station network for each epoch. Similarly, a typical PWV difference between
ECMWF PWV and GNSS PWV at the CORS station locations at UTC12:00 on DOY





214 is presented in Figure 5 (left), with the difference of the fusion model shown in
Figure 5 (right). The example in Fig 5 shows that the PWV data derived from the fusion
model are more consistent with the PWV solved by GNSS.

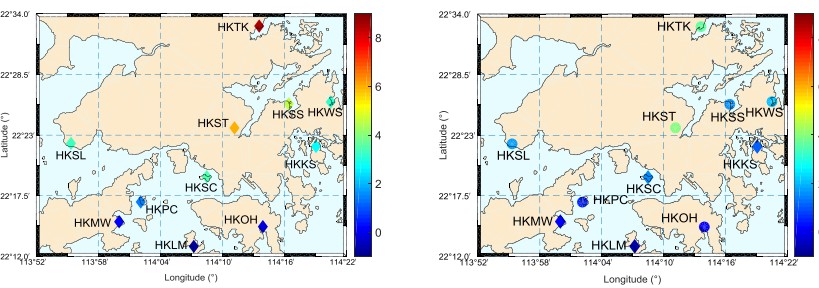


**Figure 5.** Differences (mm) between PWVs by ECMWF (left) and fusion model (right)
versus GNSS processing on the day 214 UTC 12:00
**Table 4** External precision of PWV fusion models versus CORS in August

| Doy | Sunshine (h) | Hour | Mean bias(mm) | RMS(mm) |
|---|---|---|---|---|
| 213 | 10.7 | 0 | 2.17 | 2.41 |
| 213 |  | 12 | 3.74 | 3.77 |
| 214 | 11.4 | 0 | 3.93 | 4.20 |
| 214 |  | 12 | 0.54 | 0.98 |
| 215 | 10.3 | 0 | 3.41 | 3.53 |
| 215 |  | 12 | 2.06 | 2.29 |
| 216 | 5.7 | 0 | 2.04 | 2.11 |
| 216 |  | 12 | -1.72 | 2.10 |
| 217 | 10.8 | 0 | 2.67 | 2.93 |
| 217 |  | 12 | -1.08 | 1.75 |
| 218 | 10.6 | 0 | 5.24 | 5.34 |
| 218 |  | 12 | 4.87 | 5.02 |
| 219 | 10.0 | 0 | 4.37 | 4.52 |
| 219 |  | 12 | 6.24 | 6.43 |
| mean |  |  | 2.75 | 3.38 |

Similar to the results in case 1, the PWV obtained from the fusion model that
includes data from 7 CORS stations has a mean bias of 2.75 mm and a RMS of 3.38
mm, so it is still considerably accurate. In addition, the accuracy of the fusion model
on CORS stations is slightly higher during calm troposphere conditions. This result
confirms that adding the plentiful GNSS water vapor information to the ECMWF
reanalysis product provides a definite improvement in the accuracy and reliability of
ECMWF PWV. If a larger amount of evenly distributed CORS network data are
incorporated into the PWV fusion model, the overall level of PWV calculation accuracy
in Hong Kong will increase further.
The accuracy of the precipitable water vapor obtained from GNSS and WVR
measurements is approximately 2 mm (Li et al., 2003). The fact that GNSS- and WVR-



derived water vapor data have higher accuracy than ECMWF water vapor data implies
that the assimilation of water vapor data observed from multiple techniques (e.g., GNSS,
WVR, and radiosonde) into the ECMWF can further enhance the ECMWF's weather
forecasting performance. As such, these measurements will be an important component
of the fusion model and will enhance the precipitable water vapor precision in
meteorological research.

## 4. Discussion

In order to study the meteorological application of PWV fusion model, the spatial–
temporal PWV variability as a function of topography and climatic differences will be
discussed in this section. The spatial resolution of PWV distribution presented in this
part is much higher than that of ECMWF data or GNSS data. For example, the PWV
distribution of ECMWF, CORS, and fusion model at DOY 201 UTC 00:00 are
displayed in the Figure 7. The fusion model can reflect the detailed PWV distribution
in the areas marked with red ellipse. However, when only looking at ECMWF data (or
only the GNSS data), the spatial feature is relatively coarse. In order to apply PWV to
meteorological analysis the PWV is calculated with the proposed PWV fusion model
at a spatial resolution of 1"×1" in longitude and latitude at 0:00 and 12:00 UTC during
(DOY) 201~207 and 213~219, respectively. To study the spatial distribution
considering the PWV variation with the terrain, all PWV values are reduced to the earth
surface. Hong Kong's topography is shown in Figure 6.

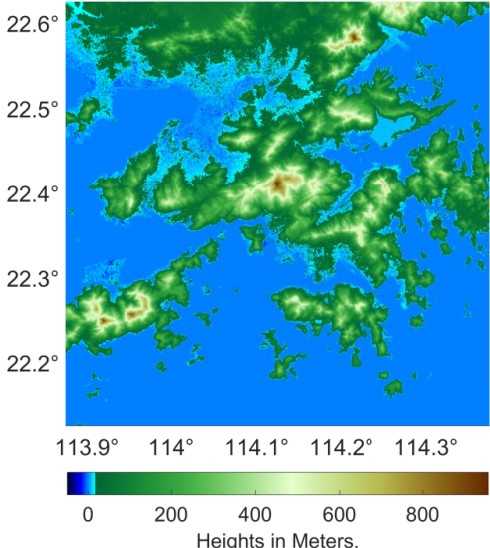


**Figure 6.** Topography in Hong Kong




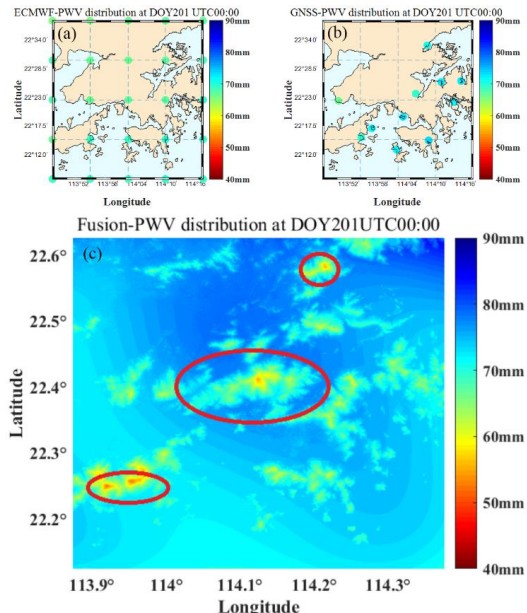


Figure 7. PWV distribution at DOY 201 UTC 00:00

Figure 8 shows the precipitable water vapor values for July, and Figure 9 shows
the PWV distribution for August.

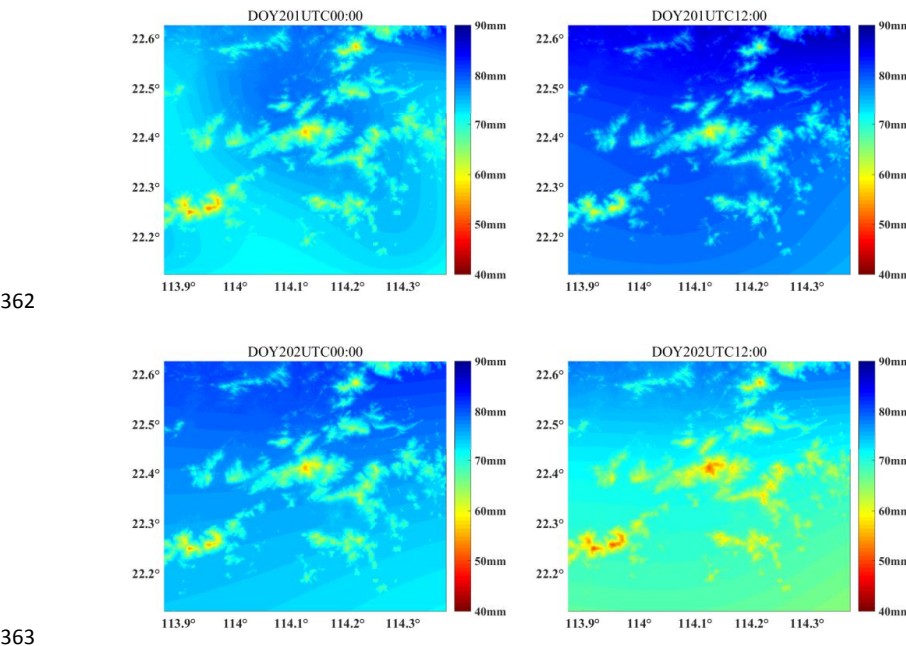













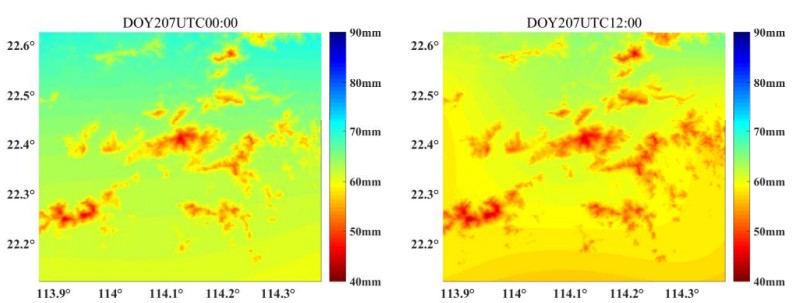


**Figure 8.** Precipitable water vapor values for July

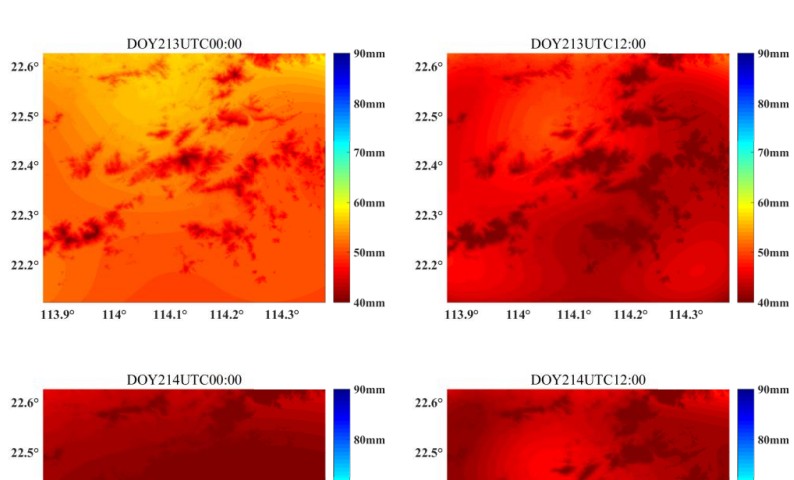



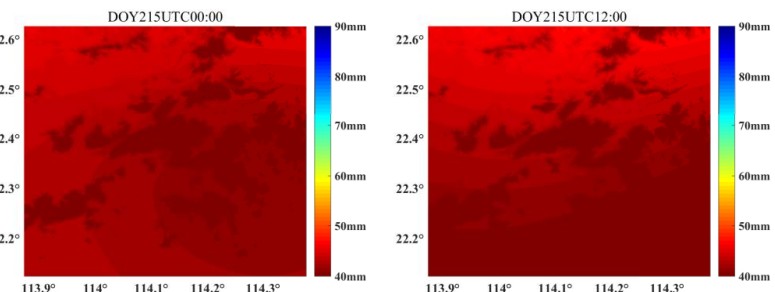











**Figure 9.** Precipitable water vapor values for August







### 4.1. Spatial variation of PWV

Figures 8 and 9 show the PWV values for the Hong Kong area. As shown in the figures, in both July and August, all the values are lower in the southern region, while the values over the northern region are relatively higher. The PWV values for DOY 201 12:00 (UTC) to DOY202 (0:00) in the northern region reach 90 mm. These relatively elevated values can be explained by the monsoon, which contrasts the results in the southern region because Hong Kong suffered a monsoon from the southern direction in July and August.

Monsoon is an important phenomenon in the Hong Kong weather context and is significant for the coastal ecosystem. The monthly and seasonal variations in precipitable water vapor are related to the onset of monsoons in the region (Joshi et al., 2013). Hong Kong is in the zone of influence of the tropical southwest and subtropical southeast monsoon in summer, and northeast monsoons are predominant in winter. There are two monsoons affecting Hong Kong during summer. First, Hong Kong is affected by the subtropical southeastern monsoon at the beginning of the pre-season flood season in early summer in southern China. With the onset of the southwest monsoon from the South China Sea, Hong Kong will gradually be affected by the more humid tropical southwest monsoon until the end of the summer monsoon. Therefore, during DOY201-207 and 213-219, the wind direction, from south to north, is similar. The monsoon carries a large amount of water vapor, making PWV generally appear to be lower in the south and higher in the north in Hong Kong.

### 4.2. Temporal variation of PWV with weather

Figures 8 and 9 show the precipitable water vapor values for different weather conditions. The rainy July with a high PWV content and the sunny August with a low PWV content are the two typical weather conditions for all regions.

The weather condition during DOY201~207 is relatively active compared with those on the preceding and following days. Several severe rainfall events occur on these days, indicating an accumulating phase of the troposphere ZWD.

Variations in precipitable water vapor correspond to the meteorological phenomena during wet and dry weather. The rainfall in Hong Kong on DOY201 is 46.2 mm, increases to 191.3 mm on DOY203 and decreases to 5.7 mm in the following days. Accordingly, a first upward and then downward trend is identified in PWV variation during that week in July. An evident cause of the high PWV values on those days is the longer period of rainfall.

In contrast, the PWV during the following sunny days of DOY213~219, however, is approximately 30 mm less than the value of the previous rainy days. On days when the sunshine duration reaches 11 h, such as DOY202, the PWV is less than 50 mm for the entire area.

These figures show that the PWV time series are affected by the variations of the rainfall on a broad scale. In more detail, the PWV time series in HKPC CORS station on DOY 203 is analyzed, accompanied by rainfall information recorded by nearby PEN meteorological station, as shown in Figure 10. It can be seen that the PWV maintained an upward trend from 0:00 to 6:00. In the meanwhile, the continuous rainfall occurred from 4:00 on. An accumulating phase of PWV can also be identified during 17:00~19:00. However, there is no rainfall event in this period. Therefore, we note that intense rainfalls are always associated with an increase in PWV, while a PWV accumulation is not necessarily accompanied by instant significant precipitation. According to the high complexity of the processes that are conducive to rainfall (Hally



et al., 2013), this result clearly confirms that rainfall is always dependent on the water
vapor content and that the accumulation of precipitable water vapor in the atmosphere
does not mean that there will always be instant rainfall.

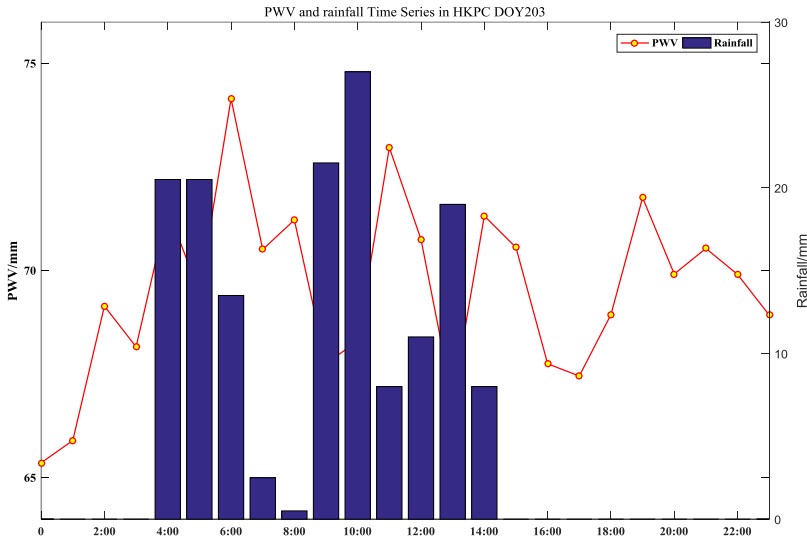


**Figure 10.** Precipitable water vapor time series and rainfall at HKPC
## 5. Conclusions
The ECMWF meteorological reanalysis product can provide precipitable water
vapor at a global scale, but ECMWF reanalysis has significant errors in the PWV field.
To improve the accuracy and reliability of the PWV estimations, this paper proposes a
local PWV fusion method that assimilates multiple sources of water vapor data through
Gaussian processes. By integrating GNSS PWV with ECMWF PWV, a precipitable
water vapor fusion model with high spatiotemporal resolution and higher precision is
established.
The proposed method has been evaluated by the Hong Kong radiosonde station
under active and quiescent troposphere conditions for DOY 2015 201~207 and
213~219. As an external reference and partial data source for modeling, 14 days of
GNSS observation data from the SatRef Network are processed by the precise point
positioning (PPP) module in the Bernese 5.0 software to establish the PWV fusion
model. With respect to radiosonde-derived PWVs, the fusion-modeled PWVs present
an accuracy of 3.87 mm in active troposphere conditions and 3.97 mm in stable
troposphere conditions, which are significantly better than the conventional ECMWF
models (i.e., 5.76 mm in active period or 6.58 mm in quiescent period). The accuracy
and spatial resolution of the PWV model have been improved to some extent by
introducing the GNSS data.
In addition, the proposed PWV fusion model is used to study the spatial–temporal
variation of precipitable water vapor over Hong Kong. Affected by the monsoon, PWV
tends to be higher in the north and lower in the south during the testing period. In

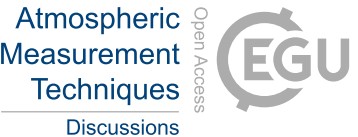

addition, the PWV values are significantly higher during the long period of rainfall than they are in fine weather.

The accuracy and reliability of the local PWV model are improved compared with those of ECMWF products. This paper has only considered the GNSS PWV data. If more sufficient data can be obtained, further efforts will also be considered to assimilate water vapor data observed via multiple techniques (e.g., GNSS, WVR) into the ECMWF reanalysis product and thereby further enhance the ECMWF's weather forecasting performance in the future.

**Acknowledgments:** The authors would like to thank Hong Kong SatRef network and ECMWF for providing experimental data. This research was supported by Key Laboratory of Geospace Environment and Geodesy, Ministry of Education, Wuhan University (16-02-03), the National Natural Science Foundation of China (41574028), the multi-dimensional water vapor inversion and assimilation based on ground-based GNSS and its application in Earth observation techniques (41704004).

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
