# Peer review of "Establishment of a regional precipitable water vapor 1 model based on the combination of GNSS and 2"

_Atmospheric Measurement Techniques, 2018_

## Referee Comment (RC1) · Anonymous Referee #1 · 23 Aug 2018

*General comments*

1. I prefer the term 'integrated water vapor' (or IWV) to 'precipitable water vapor' although I know that various authors use the latter. In this context the word precipitable seems to be either meaningless or confusing (a reader new to the subject might think there is some split between precipitable and non-precipitable water vapor).

2. What is the purpose of the manuscript: to present an IWV product for Hong Kong? There is no discussion of who the users of such a product would be. End users are much more likely to be interested in clouds or precipitation. There is a complex link between IWV and rainfall, discussed briefly in section 4.2 but I didn't feel that I learnt

anything new on the subject.

3. Another possible purpose is to persuade ECMWF (and others) to assimilate the GNSS data and hence improve analyses and forecasts (this is the gist of the last paragraph of the Conclusions). The authors cite various papers about the use of GNSS in mesoscale numerical weather prediction (NWP) systems but nothing about the use in larger scale or global NWP. They do not address the problem caused by the integrated nature of the measurement - a total increment has to be split up into a vertical profile of humidity increments and how this is done is important and far from trivial. I know that ECMWF has trialled assimilation of ground-based GNSS data (in the form of time delays) but the results were slightly disappointing and the data are not assimilated operationally for now, or in ERA5 (the replacement for ERA-Interim). Of course there is scope for improving humidity analyses and forecasts and GNSS data may well be part of that (along with improvements to the forecast model, and to the ensemble system indicating the likely structure of forecast humidity errors).

Some useful references on these subjects:

Andersson, E., Hólm, E., Bauer, P., Beljaars, A., Kelly, G. A., McNally, A. P., Simmons, A. J., Thépaut, J. and Tompkins, A. M. (2007), Analysis and forecast impact of the main humidity observing systems. Q.J.R. Meteorol. Soc., 133: 1473-1485. doi:10.1002/qj.112 [There have been various new satellites since then and also humidity sensors on some commercial aircraft.]

Poli P, Moll P, Rabier F, Desroziers G, Chapnik B, Berre L, Healy SB, Andersson E, El Guelai F-Z. 2007. Forecast impact studies of zenith total delay data from European near real-time GPS stations in Météo-France 4DVAR. J. Geophys. Res. 112: D06114, doi: 10.1029/2006JD007430.

Bennitt, G. V., A. Jupp, 2012: Operational Assimilation of GPS Zenith Total Delay Observations into the Met Office Numerical Weather Prediction Models. Mon. Wea. Rev., 140, 2706-2719. [Mainly about the regional model. I understand that the Met Office

global NWP system does use the GPS ZTD data, but the effect on the forecast is small.]

Ingleby N. B., A. C. Lorenc, K. Ngan, F. Rawlins and D. R. Jackson, 2012: Improved variational analyses using a nonlinear humidity control variable, Quarterly Journal of the Royal Meteorological Society, 139, 676, (1875-1887) [Discusses problems with vertically integrated humidity measurements.]

4. Due to the problems mentioned in comments 2 and 3 above I recommend this manuscript for rejection (and it isn't really about measurement techniques). Because of this I will only give brief comments on a few more detailed aspects.

*Detailed comments*

Title: 'Establishment' seems to imply an on-going product but there is no mention of this in the text. 'local' would be more appropriate than 'regional' (and is used at various points in the text).

Abstract:

line 10: "Water vapor is the engine of the weather." Overstatement (temperature gradients are more important, although in the tropics humidity plays a more major role than in the extratropics).

12-14 "Many techniques ... water vapor ...". Putting GNSS as number two in the list is misleading. Satellite soundings (microwave and IR) are most important, followed by radiosondes and then other sources (see Andersson et al, above). "water vapor radiometer" - it isn't clear if this relates to satellite or surface radiometers (surface radiometers are too few in number to make much difference globally and would require substantial work to assimilate the data in the best way).

15 It reads almost as if "ECMWF data" is another humidity measurement rather than a synthesis of many sources of information.

Introduction:

In several places, notably lines 48-49 and 61-62, there are "shopping lists" of references (with few after 2010). Selecting fewer might be better.

51-53 "ECMWF ... 4 times a day". There is nothing fundamental about 4 times a day, it is simply the archiving frequency chosen for ERA-Interim.

54 "the consistency and homogenous spatial coverage of ECMWF data" The homogenous coverage is a major advantage for some users, but this does not mean that the quality is homogenous (this depends on observation coverage, the synoptic situation etc). Here and in many other places it would be better to refer to "ERA-Interim data" rather than "ECMWF data".

Results:

223 "inconsistent locations" - this is a fact of life dealing with any gridded product.

234 "25 ECMWF grids": "25 ERA-Interim grid points" better (I am not sure if the grid used for archiving is the same as that used by the forecast).

Discussion:

Figures 8 and 9. I didn't feel that I learnt anything from all these plots. Just giving DOY (day of year) puts more work on the reader if they want to relate the plots to the usual calendar.

The bias between ERA-Interim and the Hong Kong radiosonde is of some interest. I would be more interested if the vertical distribution of the bias was shown (ideally for different seasons).
* * *

---

## Referee Comment (RC2) · Anonymous Referee #2 · 30 Aug 2018

**Review of "Establishment of a regional precipitable water vapor model based on the combination of GNSS and ECMWF data", by Yao, Xu, and Hu**

**General comments**

This article outlines a method for combination of ECMWF re-analysis estimates of integrated water vapour (IWV) over Hong Kong with IWVs derived from GNSS zenith total delay (ZTD) data in combination with auxiliary data from meteorological sensors installed at the GNSS sites.

It is not clear what the goal is. Is it to enhance knowledge about historic IWV values over Hong Kong above what is available from the ECMWF re-analysis product? Is it to set up a system capable of producing current IWV maps for usage in operational meteorology? Who are the users, how will the new product help them? There are some unclear points regarding the combination and the verification.

For this reason I recommend rejection of the current manuscript, but encourage the authors to make a revision and re-submit.

**Some aspects to consider in a revision**

The authors refer only to use of GNSS ZTD in regional NWP. However, GNSS ZTDs are certainly, since many years, assimilated into the NWP global models at Meteo France and the UK Met Office. I'm almost certain the situation is the same at NCEP, Environment Canada and the Japan Meteorological Office. (In the Met Office global NWP the total impact from ground-based GNSS ZTD is modest compared to several other observation systems, but the impact per GNSS ZTD observation is very large compared to most other observation types. This indicates the observations are very usefull for the NWP model, but the number of available observations is at present low compared to the number of other observations. It would be nice if the GNSS ZTDs from Hong Kong were added to the ZTDs shared internationally for NWP assimilation.)

In some equations appear variables not defined in the text.

What is meant by "earth surface"? (line 168 for example). Are you converting from the ECMWF NWP surface to the real altitude at that geographical location, or to mean sea level?

You are combining two different sources of IWV. They have different error characteristics with respect to the true IWV. When combining them the weight given to each of them is important for the end result. What determines the weight given to each of them?

You use the radiosonde (RS) and GNSS site CORS for validation. In doing so you use a specific formula to correct for altitude differences between the IWV model gridpoints and the location of the RS and CORS sites. I expect this formula to perform less well than average for mountainous regions on days with significant orographic forcing. In addition the IWV field might vary over Hong Kong due to the weather pattern, even if Hong Kong was flat. For these reasons I recommend to do a more detailed validation, in order to see if the distance to the RS and CORS has an impact on the score, likewise whether the altitude difference between the RS and CORS versus the model points has an impact.

Many people have looked for relations between IWV and precipitation, for example in the hope of making a nowcasting tool. However, it takes both water vapour and cooling (e.g., caused by buoyancy

or orographic forcing) to create precipitation, the amount of cooling necessary in part depending on relative humidity, making purely IWV based approaches in general not sucessfull. This does not rule out exceptions, that for a certain specific region one might be lucky and identify a special pattern in the IWV time variations that often relates to subsequent precipitation. However, that would require looking at a much longer time period than in this study.

Regarding potential users: The user is not likely to be an institution running an NWP model. In NWP one prefers to assimilate data in their original format. In other words assimilate GNSS ZTD directly, not an IWV value found by combining GNSS data with other data. If there is a barometer at the GNSS site, one would assimilate also that pressure observation, not use it to convert GNSS ZTD to something else.

**Detailed comment**
The driver of the weather is the sun. But don't speak about an engine. Just say water vapour is important because of its direct relation to precipitation, its role as a transport agent of energy, and its role as a green house gas. Three good causes to get more observations of water vapour.

---

## Author Comment (AC1) · 26 Oct 2018

**Response to Referee #1**

We thank the referee #1 for the insightful comments and constructive suggestions. We have addressed all their comments in the revised manuscript. Below are our response to the referee's critical comments (*Italics*).

*General comments*

*1. I prefer the term 'integrated water vapor' (or IWV) to 'precipitable water vapor' although I know that various authors use the latter. In this context the word precipitable seems to be either meaningless or confusing (a reader new to the subject might think there is some split between precipitable and non-precipitable water vapor)*

**Authors:** Thank you for your consideration. The term 'integrated water vapor' (or IWV) may be widely used in meteorological analysis. While the determination of Precipitable Water Vapor, PWV, is one of the basic objectives of GPS meteorology. Precipitable water vapor (PWV) is the total atmospheric water vapor contained in a vertical column of cross-section unit (King et al, 1992; Ichoku et al, 2002a). One way to monitor water vapour is through measurements of precipitable water vapor (PWV) using a variety of instruments onboard different platforms. Since this paper aims to obtain PWV field with higher accuracy through GPS signals using the precipitable water vapor (PWV) observation, the term precipitable water vapor (PWV) is used in this manuscript. The total precipitable water vapor is atmospheric water vapor contained in a column of unit cross section extending all of the way from the earth's surface to the "top" of the atmosphere. And this explanation has been added in the revised manuscript.

*2. What is the purpose of the manuscript: to present an IWV product for Hong Kong? There is no discussion of who the users of such a product would be. End users are much more likely to be interested in clouds or precipitation. There is a complex link between IWV and rainfall, discussed briefly in section 4.2 but I didn't feel that I learnt anything new on the subject.*

**Authors:** Thank you for your question. We are sorry for the unclear purpose of discussion in the article. On the one hand, accurate measurement of water vapor is vital for improving the predictability of regional precipitation, weather and visibility. With advances of GPS technology and spreading the GPS network around the globe, it is a great challenge to explore the application of GPS such as for rainfall, which is a cost-effective and low maintenance cost for a satellite tracking solution. The results from rainfall inferred from GPS can be used to improve nowcasting and weather prediction.The analysis of spatial and temporal variation of PWV in section 4 is to explore the application potential of PWV fusion models in meteorological analysis. ECMWF data are the main source of data for meteorological research. However, the spatial resolution of PWV distribution presented in this part is much higher than that of

ECMWF data or GNSS data.

On the other hand, the precision of PWV derived by fusion model is improved compared with commonly used ECMWF PWV data. This high-precision and high-resolution PWV data is more distinct for analyzing weather changes in a region. As mentioned in the article, the feature of north-south difference and its variation with precipitation can be clearly obtained. The application in climatology of PWV fusion model is the focus of section 4. Since the analysis of link between PWV and rainfall is in section 4.2 is too simple, we removed this part from the paper. Besides, PWV maps can also be used for water vapor correction in InSAR. With more precise PWV, the accuracy of the InSAR measurement will be improved.

Therefore, GNSS and ECMWF can be combined to obtain better-precision regional PWV products, which can not only serve for geodetic techniques, but also provide more accurate data for meteorological research.

*3. Another possible purpose is to persuade ECMWF (and others) to assimilate the GNSS data and hence improve analyses and forecasts (this is the gist of the last paragraph of the Conclusions). The authors cite various papers about the use of GNSS in mesoscale numerical weather prediction (NWP) systems but nothing about the use in larger scale or global NWP. They do not address the problem caused by the integrated nature of the measurement - a total increment has to be split up into a vertical profile of humidity increments and how this is done is important and far from trivial. I know that ECMWF has trialled assimilation of ground-based GNSS data (in the form of time delays) but the results were slightly disappointing and the data are not assimilated operationally for now, or in ERA5 (the replacement for ERA-Interim). Of course there is scope for improving humidity analyses and forecasts and GNSS data may well be part of that (along with improvements to the forecast model, and to the ensemble system indicating the likely structure of forecast humidity errors)*

**Authors:** Thank you for the suggested useful references, as shown in the articles, GNSS ZTDs are, since many years, assimilated into the NWP global models at Meteo France and the UK Met Office. And the impact of assimilating ZTD observations in numerical weather prediction (NWP) models has previously been described by authors such as Yan et al. (2009), Boniface et al. (2009), Macpherson et al. (2008), Poli et al. (2007), Faccani et al. (2005), and Vedel and Huang (2004). These paper have been cited in the revised manuscript. Although they found no improvement in temperature, wind, and humidity forecasts, these experiments indicatedpositive impact on improvement in the prediction of precipitation patterns in cases where ZTDs were assimilated. The fusion model established in our paper is local model and only for PWV, there are no other meteorological elements. According to *Yao Y, Shan L, Zhao Q. Establishing a method of short-term rainfall forecasting based on GNSS-derived PWV and its application [J]. Sci Rep, 2017, 7(1):12465*, it is feasible to predict rainfall based on single factor of PWV. Considering the fact that ECMWF products currently have not assimilated ground GNSS data, our paper provides an idea of integrating GNSS data to improve PWV accuracy issued by ECMWF, other meteorological elements are not involved.

*Detailed comments*

*Title: 'Establishment' seems to imply an on-going product but there is no mention of this in the text. 'local' would be more appropriate than 'regional' (and is used at various points in the text).*

**Authors:** We replaced the word 'regional' by 'local' in the title as you suggested and changed the title to "Local precipitable water vapor products based on the combination of GNSS and ECMWF data"

*line 10: "Water vapor is the engine of the weather." Overstatement (temperature gradients are more important, although in the tropics humidity plays a more major role than in the extratropics)*

**Authors:** We are sorry for the overstatement. Water vapour is important because of its direct relation to precipitation, its role as a transport agent of energy, and its role as a greenhouse gas. We have changed the sentence into "water vapor is an important factor of the weather" in the revised manuscript.

*12-14 "Many techniques ... water vapor ...". Putting GNSS as number two in the list is misleading. Satellite soundings (microwave and IR) are most important, followed by radiosondes and then other sources (see Andersson et al, above). "water vapor radiometer" - it isn't clear if this relates to satellite or surface radiometers (surface radiometers are too few in number to make much difference globally and would require substantial work to assimilate the data in the best way).*

**Authors:** Thank you for your advice, we have adjusted the introduction of water vapor measurements in the revised manuscript. "Many techniques are used for measuring the water vapor in the atmosphere such as satellite soundings (microwave and InfraRed), radiosondes, Global Navigation Satellite System (GNSS) and satellite water vapor radiometer (WVR)."

*15 It reads almost as if "ECMWF data" is another humidity measurement rather than a synthesis of many sources of information.*

**Authors:** We have added the explanation about ECMWF data as you suggested. "ECMWF currently provides ERA-Interim reanalysis data, which incorporates many important IFS improvements such as model resolution and physics changes, the use of four-dimensional variational data assimilation (4-D-Var), and various other changes in the analysis methodology (Dee et al., 2011)."

*Introduction:*
*In several places, notably lines 48-49 and 61-62, there are "shopping lists" of references (with few after 2010). Selecting fewer might be better.*

**Authors:** Thank you for your advice, we have deleted some references in the corresponding part.

*51-53 "ECMWF ... 4 times a day". There is nothing fundamental about 4 times a day, it*

*is simply the archiving frequency chosen for ERA-Interim.*
**Authors:** We have revised the detailed time resolution about ERA-Interim.

.

*54 "the consistency and homogenous spatial coverage of ECMWF data" The homogenous coverage is a major advantage for some users, but this does not mean that the quality is homogenous (this depends on observation coverage, the synoptic situation etc). Here and in many other places it would be better to refer to "ERA-Interim data" rather than "ECMWF data"*
**Authors:** We admit that the quality of ECMWF data is necessarily homogenous. Compared with other data sources, ECMWF data is global and consistent, in the sense that its different fields are in some balance. We replaced some "ECMWF data" by "ERA-Interim data" throughout the paper as you suggested

*Results:*
*223 "inconsistent locations" - this is a fact of life dealing with any gridded product*
**Authors:** We agree that interpolation cannot be avoided in the process of using grid data

*234 "25 ECMWF grids": "25 ERA-Interim grid points" better (I am not sure if the grid used for archiving is the same as that used by the forecast).*
**Authors:** Thank you for your advice, we replaced some "25 ECMWF grids" by "25 ERA-Interim grid points" throughout the paper as you suggested

*Discussion:*
*Figures 8 and 9. I didn't feel that I learnt anything from all these plots. Just giving DOY (day of year) puts more work on the reader if they want to relate the plots to the usual calendar.*
**Authors:** There two figures show the PWV values for the Hong Kong area during two typical weather condition. It set up a system capable of producing current PWV maps for usage in operational meteorology. Thank you for the helpful advice, we have replotted the plots as the reviewer suggested.

*The bias between ERA-Interim and the Hong Kong radiosonde is of some interest. I would be more interested if the vertical distribution of the bias was shown (ideally for different seasons).*
**Authors:** In fact, the PWV derived from the fusion model in this paper is a two-dimensional product without vertical structure, which provide the PWV value at earth surface height.

---

## Author Comment (AC2) · 26 Oct 2018

**Response to Referee #2**

We thank the referee #2 for the insightful comments and constructive suggestions. We have addressed all their comments in the revised manuscript. Below are our response to the referee's critical comments (*Italics*).

**General comments**

*This article outlines a method for combination of ECMWF re-analysis estimates of integrated water vapour (IWV) over Hong Kong with IWVs derived from GNSS zenith total delay (ZTD) data in combination with auxiliary data from meteorological sensors installed at the GNSS sites.*

*It is not clear what the goal is. Is it to enhance knowledge about historic IWV values over Hong Kong above what is available from the ECMWF re-analysis product? Is it to set up a system capable of producing current IWV maps for usage in operational meteorology? Who are the users, how will the new product help them? There are some unclear points regarding the combination and the verification.*
*For this reason I recommend rejection of the current manuscript, but encourage the authors to make a revision and re-submit.*

**Authors:** We are sorry that the purpose of the proposed PWV fusion model is not explained clearly. According to our investigation of various PWV measurements, ECMWF reanalysis product ERA-Interim is a commonly used data source in meteorological research. However, ECMWF reanalysis has significant errors in the PWV field. In the meanwhile, the GNSS derived PWV, which is only available in CORS stations, is proved to be more accurate. Therefore, the merge of ECMWF and GNSS data is to obtain PWV data with higher accuracy than ECMWF and relatively higher horizontal resolution as well. The aim of fusion model lies on the precision improvement of ECMWF PWV and further exploration about its application potential in geodesy and climatology. The purpose of the study is more specifically emphasized in the revised manuscript at line 84-85.

On the one hand, accurate measurement of water vapor is vital for improving the predictability of regional precipitation, weather and visibility. With advances of GPS technology and spreading the GPS network around the globe, it is a great challenge to explore the application of GPS such as for rainfall, which is a cost-effective and low maintenance cost for a satellite tracking solution. The results from rainfall inferred from GPS observations can be used to improve nowcasting and weather prediction. So the analysis of spatial and temporal variation of PWV in section 4 is to explore the application potential of PWV fusion models in meteorological analysis. As we know, ECMWF data are the main source of data for meteorological research. However, the spatial resolution of PWV distribution presented in this part is much higher than that of ECMWF data or GNSS data.

On the other hand, the precision of PWV derived by fusion model is improved

compared with commonly used ECMWF PWV data. This high-precision and high-resolution PWV data is more distinct for analyzing weather changes in a region. As mentioned in the article, the feature of north-south difference and its variation with precipitation can be clearly obtained. The application in climatology of PWV fusion model is the focus of section 4. The fusion model can reflect the detailed PWV distribution, which could provide more accurate data for meteorologists to analyze. As a result, full use of high-precision, high-resolution PWV data obtained from fusion model is important for meteorological research. Besides, PWV maps can also be used for water vapor correction in InSAR. With more precise PWV, the accuracy of the InSAR measurement will be improved.

Therefore, GNSS and ECMWF can be combined to obtain better-precision regional PWV products, which can not only serve for geodetic techniques, but also provide more accurate data for meteorological research.

*Some aspects to consider in a revision*
*The authors refer only to use of GNSS ZTD in regional NWP. However, GNSS ZTDs are certainly, since many years, assimilated into the NWP global models at Meteo France and the UK Met Office. I'm almost certain the situation is the same at NCEP, Environment Canada and the Japan Meteorological Office. (In the Met Office global NWP the total impact from ground-based GNSS ZTD is modest compared to several other observation systems, but the impact per GNSS ZTD observation is very large compared to most other observation types. This indicates the observations are very useful for the NWP model, but the number of available observations is at present low compared to the number of other observations. It would be nice if the GNSS ZTDs from Hong Kong were added to the ZTDs shared internationally for NWP assimilation.)*

**Authors:** The impact of assimilating ZTD observations in numerical weather prediction (NWP) models has previously been described by authors such as Yan et al. (2009), Boniface et al. (2009), Macpherson et al. (2008), Poli et al. (2007), Faccani et al. (2005), and Vedel and Huang (2004). Although they found no improvement in temperature, wind, and humidity forecasts, these experiments saw positive impact on improvement in the prediction of precipitation patterns in cases where ZTDs were assimilated. As we know, GPS network is spreading around the globe, and Hong Kong is a dense area of GPS network. Accurate measurement of water vapor is vital for improving the predictability of regional precipitation, weather and visibility, especially for a highly moist metropolis such as Hong Kong with strong weather variability and the unique characteristic metorology (Chen and Liu, 2014). Therefore, it is very necessary to make full use of the available GNSS data here to enhance meteorological observations. The experiment in this paper shows that the fusion of GNSS data and ERA-Interim reanalysis data can significantly improve the accuracy of PWV.

*In some equations appear variables not defined in the text.*
**Authors:** Thank you for your reminding, we have added the definition of the variables in each equations.

*What is meant by "earth surface"? (line 168 for example). Are you converting from the ECMWF NWP surface to the real altitude at that geographical location, or to mean sea level?*

**Authors:** In this paper, we convert from the ECMWF NWP surface to the real altitude at that geographical location, the PWV derived from the fusion model is a two-dimensional product, providing the PWV value at earth surface height.

*You are combining two different sources of IWV. They have different error characteristics with respect to the true IWV. When combining them the weight given to each of them is important for the end result. What determines the weight given to each of them?*

**Authors:** According to the formula (5), the term $\varepsilon$ accounts for the error, and is assumed to follow a normal distribution with 0 mean and $\sigma_\varepsilon^2$ variance, i.e. $\varepsilon \sim N(0, \sigma_\varepsilon^2)$. In the ideal case of a perfect fitting, that is when the model $f(v_i)$ of the underlying surface is appropriate and the related coefficients are correctly estimated, $\varepsilon$ should represent the measurement error. Since ECMWF data lack the information about the variable uncertainties, for simplicity we used one parameter $\sigma_\varepsilon^2$ for the GNSS and ECMWF data in Gaussian Process model.

*You use the radiosonde (RS) and GNSS site CORS for validation. In doing so you use a specific formula to correct for altitude differences between the IWV model gridpoints and the location of the RS and CORS sites. I expect this formula to perform less well than average for mountainous regions on days with significant orographic forcing. In addition the IWV field might vary over Hong Kong due to the weather pattern, even if Hong Kong was flat. For these reasons I recommend to do a more detailed validation, in order to see if the distance to the RS and CORS has an impact on the score, likewise whether the altitude difference between the RS and CORS versus the model points has an impact.*

**Authors:** The PWV involved in modelling have been height (m)-reduced to the earth surface using coefficients obtained from Ref (Means, 2011). And the height should not exceed 1000m in this formula). The height of CORS stations and Radiosonde stations in the Hong Kong area is generally below 1000m. We believe that the comparison between PWV values of the RS and CORS stations after height conduction is more accurate.

*Many people have looked for relations between IWV and precipitation, for example in the hope of making a nowcasting tool. However, it takes both water vapour and cooling (e.g., caused by buoyancy or orographic forcing) to create precipitation, the amount of cooling necessary in part depending on relative humidity, making purely IWV based approaches in general not sucessfull. This does not rule out exceptions, that for a certain specific region one might be lucky and identify a special pattern in the IWV time*

*variations that often relates to subsequent precipitation. However, that would require looking at a much longer time period than in this study。*

**Authors:** According to *Yao Y, Shan L, Zhao Q. Establishing a method of short-term rainfall forecasting based on GNSS-derived PWV and its application [J]. Sci Rep, 2017, 7(1):12465*, it is feasible to predict rainfall based on single factor of PWV. Considering the fact that ECMWF products currently have not assimilated ground GNSS data, our paper provides an idea of integrating GNSS data to improve PWV accuracy issued by ECMWF, other meteorological elements are not involved.

Thank you for your suggestion, we are sorry that the reason for the period used for the evaluation in this paper is not explained clearly. The primary purpose of the merge of ECMWF and GNSS data is to obtain PWV data with higher accuracy than ECMWF. The typical weather experiment in two weeks shows that the precision improvement is significant. From the perspective of fusion model establishment, there are not much difference in accuracy whether the period used is two weeks or several years. As far as we concerned, the main meteorological impacts on PWV are rain and no rain. Therefore the period we select include two typical weather conditions, which have influence on the PWV variation.

The aim of fusion model lies on the precision improvement of ECMWF PWV. Further discussion in climatology is an additional exploration about the application potential of fusion model. The PWV response to simple weather are therefore analyzed. If large-scale climate change is to be studied, the long-term GNSS observations and ECMWF data should be considered in PWV fusion model. So far this paper only put emphasis on proving the feasibility of PWV merge to improve accuracy using Gaussian Process method. Thank you for your useful suggestion, we will select a longer period of time in the study of PWV and precipitation.

*Regarding potential users: The user is not likely to be an institution running an NWP model. In NWP one prefers to assimilate data in their original format. In other words assimilate GNSS ZTD directly, not an IWV value found by combining GNSS data with other data. If there is a barometer at the GNSS site, one would assimilate also that pressure observation, not use it to convert GNSS ZTD to something else.*

**Authors:** There are plenty of research about the GNSS ZTD assimilation in NWP, and to persuade ECMWF (and others) to assimilate the GNSS data is not the Not the main purpose of this article, we target at the single factor PWV.

We have to identify that the user of the fusion model might be meteorologist. As we know, ECMWF is an important resource for conducting meteorological research. With the fusion model established in our paper, we can obtain PWV data with higher accuracy than ECMWF, providing more precise and detailed information for meteorological analysis.

In addition, our paper provides an idea of integrating GNSS data to improve PWV accuracy issued by ECMWF, other meteorological elements are not involved.Another users might be geodetic surveying users, because PWV maps can be used for water vapor correction in InSAR. With more precise PWV, the accuracy of the InSAR measurement will be improved.

*Detailed comment*

*The driver of the weather is the sun. But don't speak about an engine. Just say water vapour is important because of its direct relation to precipitation, its role as a transport agent of energy, and its role as a green house gas. Three good causes to get more observations of water vapour.*

**Authors:** We are sorry for the overstatement. Water vapor is an important component of the atmosphere, which affects the radiation balance, energy transportation, and the formation of cloud and precipitation. We have changed the description in the revised manuscript.